# VOGUE: Guiding Exploration with Visual Uncertainty Improves Multimodal Reasoning

## Abstract

Reinforcement learning with verifiable rewards (RLVR) improves reasoning in LLMs but struggles with exploration, an issue that still persists for Multimodal LLMs (MLLMs). Current methods treat the visual input as a fixed, deterministic condition, overlooking a critical source of ambiguity and struggling to build policies robust to plausible visual variations. We introduce **VOGUE** (**Visual-Uncertainty–Guided Exploration**), a novel method that shifts exploration from the output (text) to the input (visual) space. By treating the image as a stochastic context, VOGUE quantifies the policy's sensitivity to visual perturbations using the symmetric KL divergence between a "raw" and "noisy" branch, creating a direct signal for uncertainty-aware exploration. This signal shapes the learning objective via an uncertainty-proportional bonus, which, combined with a token-entropy bonus and an annealed sampling schedule, effectively balances exploration and exploitation. Implemented within GRPO on two model scales (Qwen2.5-VL-3B/7B), VOGUE boosts pass@1 accuracy by an average of 2.6% on three visual math benchmarks and 3.7% on three general-domain reasoning benchmarks, while simultaneously increasing pass@4 performance and mitigating the exploration decay commonly observed in RL fine-tuning. Our work shows that grounding exploration in the inherent uncertainty of visual inputs is an effective strategy for improving multimodal reasoning.

## 1 Introduction

Reinforcement learning with verifiable rewards (RLVR) has substantially improved the reasoning abilities of large language models (LLMs) by optimizing against ground-truth answers (Luong et al., 2024; Lambert et al., 2024; Guo et al., 2025; Su et al., 2025). However, this outcome-centric approach often biases learning toward exploitation, suppressing trajectories with valid intermediate reasoning that conclude with an incorrect answer. This limitation stifles exploration and can lead to brittle policies (Dai et al., 2025). While this challenge is recognized in text-only domains, with mitigation strategies including uncertainty-aware objectives (Cheng et al., 2025), diversity-promoting rewards (Li et al., 2025a), pass@k rewards (Chen et al., 2025b; Walder and Karkhanis, 2025), and intermediate feedback (Setlur et al., 2024), these methods do not address the unique sources of uncertainty inherent to multimodal reasoning.

This exploration problem is arguably amplified in Multimodal LLMs (MLLMs), where textual reasoning is grounded in complex visual inputs (Huang et al., 2025; Tan et al., 2025; Peng et al., 2025). Current multimodal RLVR approaches typically treat the image as a fixed, deterministic condition. This overlooks a key source of ambiguity: the visual modality itself. An image can contain ambiguous objects, be subject to multiple valid interpretations, or have its crucial details altered by plausible perturbations. By not probing these visual uncertainties, existing methods do not explicitly incentivize policies to test the robustness of their visual understanding. Consequently, models may learn spurious visual-text correlations rather than developing deep, generalizable reasoning, leaving a critical question unanswered: **How can we leverage visual uncertainty to drive more effective exploration?**

To address this gap, we introduce **Visual-Uncertainty–Guided Exploration** (**VOGUE**), a novel method that makes exploration modality-aware. As illustrated in Figure 1, VOGUE shifts exploration from the output (text) space to the input (visual) space by treating the image as a stochastic

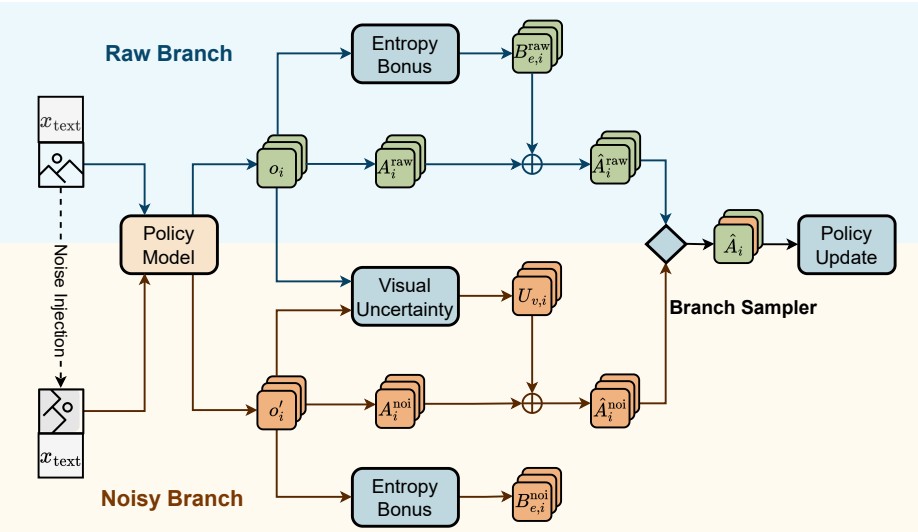

Figure 1: **VOGUE for RL fine-tuning**. Our method uses a dual-branch forward pass: the raw branch processes the original image, while the noisy branch receives a perturbed view. Token-level symmetric KL between branches provides a visual-uncertainty signal used to shape the noisy-branch advantage. An entropy bonus on both branches maintains output stochasticity, and an annealed sampling schedule balances exploration and exploitation by favoring the noisy branch early in training.

context. For each training example, we perform a dual-branch forward pass: one for a "raw" branch using the original image and another for a "noisy" branch using a semantics-preserving perturbed view. We quantify visual uncertainty as the symmetric KL divergence between the policy distributions induced by these two views. This identifies states where the model's predictions are brittle to plausible visual perturbations and are therefore states worthy of exploration. We then shape advantages with (i) a capped, uncertainty-proportional visual uncertainty bonus on the noisy branch to focus exploration on visually ambiguous inputs, and (ii) a token-entropy bonus on both branches to maintain policy stochasticity. To manage the exploration–exploitation trade-off, we employ an annealed branch-sampling schedule that prioritizes uncertainty-driven exploration early in training before shifting focus to the original view as learning stabilizes. In essence, VOGUE couples textual exploration with a measure of visual confidence, pushing the model to resolve ambiguities and build more robust reasoning skills.

We implement VOGUE within GRPO (Shao et al., 2024) and evaluate it on six diverse mathematical and general-domain reasoning benchmarks: MathVerse (Zhang et al., 2024), MathVista (Lu et al., 2023), WeMath (Qiao et al., 2024), HallusionBench (Guan et al., 2024), ChartQA (Masry et al., 2022), and LogicVista (Xiao et al., 2024). On Qwen2.5-VL-3B and 7B models (Bai et al., 2023) trained on MMRL30k (Zhu et al., 2025a), VOGUE delivers substantial improvements over strong baselines at both model scales, boosting pass@1 accuracy by an average of 2.6% on three visual math benchmarks and 3.7% on three general-domain reasoning benchmarks (Table 1 and Table 2). Crucially, VOGUE also increases pass@4 performance, effectively mitigating the exploration decay often seen in RL fine-tuning, a challenge that methods like GRPO face (Table 3). Furthermore, VOGUE consistently outperforms Pass@k Training, a dedicated exploration-promoting method primarily demonstrated effective in text-only settings, yielding both higher pass@1 and more consistent pass@k gains. In summary, our contributions are three-fold:

- We identify visual uncertainty as a key, yet overlooked, mechanism for exploration in MLLMs and propose to leverage it to improve MLLM reasoning.

- We introduce VOGUE, a practical method that uses a dual-branch architecture to quantify visual uncertainty, a capped, uncertainty-proportional advantage bonus, and an annealed sampling schedule.

- We provide extensive empirical validation demonstrating that VOGUE consistently improves both **exploitation** (pass@1) and **exploration** (pass@k) over strong baselines.

## 2 PRELIMINARIES

We adopt GRPO (Shao et al., 2024) as the underlying RL algorithm in this work. In GRPO, given an input $x$, a group of responses $\{o_i\}_{i=1}^G$ are sampled from the old policy $\pi_{\theta_{\text{old}}}$, each associated with a reward $r_i$. Then the normalized advantage for response $o_i$ is defined as:

$$A_i = \frac{r_i - \text{mean}(\{r_i\}_{i=1}^G)}{\text{std}(\{r_i\}_{i=1}^G)}. \tag{1}$$

As in PPO, GRPO uses clipped importance sampling to stabilize policy updates. Let $\rho_i(\theta) = \frac{\pi_\theta(o_i|x)}{\pi_{\theta_{\text{old}}}(o_i|x)}$ denote the probability ratio between the new and old policies. The GRPO objective is to maximize the following:

$$\mathcal{J}_{\text{GRPO}}(\theta) = \mathbb{E}_{x \sim \mathcal{D}, \{o_i\} \sim \pi_{\theta_{\text{old}}}(\cdot|x)} \left[ \frac{1}{G} \sum_{i=1}^G \min\left(\rho_i(\theta)A_i, \; \text{clip}\left(\rho_i(\theta), 1 - \epsilon_{\text{clip}}, 1 + \epsilon_{\text{clip}}\right)A_i\right) \right], \tag{2}$$

where $\epsilon_{\text{clip}}$ is the clipping hyperparameter.

## 3 VISUAL UNCERTAINTY-GUIDED EXPLORATION

To encourage exploration in multimodal RLVR, we propose **visual-uncertainty–guided exploration (VOGUE)**. Formally, with input $x = (x_{\text{text}}, x_{\text{image}})$, we aim to optimize an MLLM policy network $\pi_\theta$ by maximizing a surrogate objective in Eq. 2. As illustrated in Figure 1, our approach employs a dual-branch forward pass and treats the image as a stochastic context. For each input, the policy is evaluated on both the raw and a semantics-preserving noisy view, and visual uncertainty is quantified via the symmetric KL divergence between the resulting text policy distributions. This visual uncertainty then guides advantage shaping and, combined with an annealed sampling schedule, steers the model to explore visually ambiguous states early while focusing on the original view as training stabilizes. The full procedure is summarized in Algorithm 1.

### 3.1 VISUAL UNCERTAINTY

The core of our approach is to enhance exploration in multimodal RLVR by shifting the focus from the output (text) space to the input (visual) space, treating the image as a stochastic context rather than a fixed condition. To this end, we introduce controlled perturbations to the visual input and define **visual uncertainty** as the extent to which the model's output distribution varies under semantics-preserving transformations of the image. Variation of the model's predictions indicate regions of the visual state space where additional exploration is likely to improve policy reasoning and robustness.

To induce these perturbations for encouraging exploration, we apply stochastic image augmentation. For each image $x_{\text{image}}$ in the training dataset, we create a perturbed counterpart $x'_{\text{image}}$ through a stochastic augmentation function $\mathcal{T}$. This function applies a composition of transformations: $x'_{\text{image}} = \mathcal{T}(x_{\text{image}})$, where $\mathcal{T}$ includes random horizontal/vertical flips, rotations, color jittering, and the addition of Gaussian noise. These augmentations are designed to preserve the core semantic content of the image while altering its low-level feature representation, ensuring that differences in the model's output reflect true sensitivity to visual variations.

Then we employ a dual-branch forward pass. The raw branch processes the original input $x = (x_{\text{text}}, x_{\text{image}})$ to produce an output probability distribution $P = \pi_\theta(\cdot|x)$, while the noisy branch uses the perturbed input $x' = (x_{\text{text}}, x'_{\text{image}})$ to produce a distribution $Q = \pi_\theta(\cdot|x')$. After obtaining the two distributions, we represent the visual uncertainty $U_v$ as the divergence between them. We measure this using a symmetric KL divergence, which is calculated as the mean of the forward and backward KL divergences, encouraging exploration while maintaining stability:

$$U_v = \frac{1}{2}\left(D_{KL}(P||Q) + D_{KL}(Q||P)\right). \tag{3}$$

### 3.2 ADVANTAGE SHAPING

We maintain separate advantage calculations for the raw and noisy branches, as shown in Figure 1. To encourage exploration, we introduce a **visual uncertainty bonus** $B_v$ to the noisy

---

**Algorithm 1** Visual-Uncertainty-Guided Exploration (VOGUE)

---

**Require:** Dataset $\mathcal{D}$, total training steps $S_{\text{total}}$, group size $G$, annealing schedule parameters $p_{\text{start}}, p_{\text{end}}$, scaling factors $\alpha_v, \beta_v, \alpha_e, \beta_e$, augmentation function $\mathcal{T}$

1: Initialize policy parameters $\theta$, old policy $\theta_{\text{old}} \leftarrow \theta$
2: **for** $s = 1$ to $S_{\text{total}}$ **do**
3:     Sample input $x = (x_{\text{text}}, x_{\text{image}}) \sim \mathcal{D}$
4:     Construct perturbed image $x'_{\text{image}} = \mathcal{T}(x_{\text{image}})$, $x' = (x_{\text{text}}, x'_{\text{image}})$
5:     Sample group of responses $\{o_i\}_{i=1}^{G} \sim \pi_{\theta_{\text{old}}}(\cdot \mid x)$, $\{o'_i\}_{i=1}^{G} \sim \pi_{\theta_{\text{old}}}(\cdot \mid x')$
6:     Compute annealed probability: $p_{\text{noi}}(s) = p_{\text{end}} + (p_{\text{start}} - p_{\text{end}}) \cdot \max\left(0, 1 - \frac{s}{S_{\text{total}}}\right)$
7:     **for** $i = 1$ to $G$ **do**
8:         Compute raw advantage $A_i^{\text{raw}}$, noisy advantage $A_i^{\text{noi}}$ (Eq. 1)
9:         Compute token entropy $H_i^{\text{raw}}$ and $H_i^{\text{noi}}$ for each branch
10:        Compute visual uncertainty $U_v$ using token-level symmetric KL divergence (Eq. 3)
11:        Compute bonuses:

$$B_v = \min\left(\frac{|A_i^{\text{noi}}|}{\beta_v}, \alpha_v \cdot \text{stopgrad}(U_v)\right)$$

$$B_e^{\text{raw}} = \min\left(\frac{|A_i^{\text{raw}}|}{\beta_e}, \alpha_e \cdot \text{stopgrad}(H_i^{\text{raw}})\right), B_e^{\text{noi}} = \min\left(\frac{|A_i^{\text{noi}}|}{\beta_e}, \alpha_e \cdot \text{stopgrad}(H_i^{\text{noi}})\right)$$

12:        Compute shaped advantages:

$$\widehat{A}_i^{\text{raw}} = A_i^{\text{raw}} + B_e^{\text{raw}}, \quad \widehat{A}_i^{\text{noi}} = A_i^{\text{noi}} + B_e^{\text{noi}} + B_v$$

13:        Sample branch selector $z_i \sim \text{Bernoulli}(p_{\text{noi}}(s))$
14:        Select final advantage (with corresponding selected $(x, o_i)$ or $(x', o'_i)$):

$$\widehat{A}_i = z_i \cdot \widehat{A}_i^{\text{noi}} + (1 - z_i) \cdot \widehat{A}_i^{\text{raw}}$$

15:     **end for**
16:     Compute surrogate objective with shaped advantages (Eq. 2)
17:     Update policy parameters $\theta \leftarrow \theta + \eta \nabla_\theta \mathcal{J}_{\text{VOGUE}}(\theta)$
18:     Periodically update old policy $\theta_{\text{old}} \leftarrow \theta$
19: **end for**

---

branch. This bonus, based on the visual uncertainty $U_v$, guides the branch to explore regions of the state space that may not be reachable by the raw branch. The bonus $B_v$ is defined as: $B_v = \min\left(\frac{|A^{\text{noi}}|}{\beta_v}, \alpha_v \cdot \text{stopgrad}(U_v)\right)$, where $A^{\text{noi}}$ is the advantage for the noisy branch, $\alpha_v$ and $\beta_v$ are scaling factors, $\text{stopgrad}(\cdot)$ is the stop gradient operator. Furthermore, to promote general policy stochasticity and exploration, we incorporate an **entropy bonus** $B_e$ for both branches. This bonus is based on the token entropy $H$ of the policy's output distribution, which is defined as: $H = -\sum_{v \in \mathcal{V}} \pi_\theta(v \mid x, o_{<t}) \log \pi_\theta(v \mid x, o_{<t})$. The entropy bonus $B_e$ is defined as: $B_e = \min\left(\frac{|A|}{\beta_e}, \alpha_e \cdot \text{stopgrad}(H)\right)$, where $\alpha_e$ and $\beta_e$ are scaling factors, and $\mathcal{V}$ denotes the vocabulary. Therefore, for the raw branch, the shaped advantage is calculated as: $\hat{A}^{\text{raw}} = A^{\text{raw}} + B_e^{\text{raw}}$, and for the noisy branch: $\hat{A}^{\text{noi}} = A^{\text{noi}} + B_e^{\text{noi}} + B_v$. When implementing VOGUE with GRPO, we use GRPO's standard estimator to compute the base advantages $A^{\text{noi}}$ and $A^{\text{raw}}$ (see Eq. 1).

The policy gradient for the noisy branch can be expressed as follows:

$$\nabla_\theta \mathcal{J}_{\text{VOGUE}}(\theta) \propto \mathbb{E}_{o' \sim \pi_{\theta_{\text{old}}}} \left[(A^{\text{noi}} + B_e^{\text{noi}} + \alpha_v U_v) \nabla_\theta \log \pi_\theta(o' \mid x')\right] \tag{4}$$

From a gradient perspective, this formulation encourages more effective exploration compared to standard RLVR approaches such as GRPO. We omit caps/clipping for clarity. The term $\alpha_v U_v \nabla_\theta \log \pi_\theta(o' \mid x')$ explicitly encourages the policy to increase the probability of action sequences that follow from visually uncertain states, guiding the model to acquire more informative visual features. There is also an entropy bonus $B_e$. This acts as a general-purpose exploration mech-

anism that complements the exploration driven by $B_v$. While $B_v$ directs exploration toward visual uncertainty, $B_e$ maintains stochasticity in the textual output space.

### 3.3 ANNEALED SAMPLING FOR POLICY OPTIMIZATION

During training, it is crucial to balance the aggressive exploration driven by the noisy branch with the stable learning provided by the raw branch. A policy trained exclusively on the noisy branch may become overly stochastic and fail to converge, while a policy trained solely on the raw branch may not explore enough to find the optimal path. To manage this trade-off, we employ an **annealed sampling strategy**. At each training step, for each sample in the batch, we stochastically choose which advantage estimate to use for the policy update. We define $p_{\text{noi}}$ as the probability of selecting the advantage from the noisy branch, which is expressed as:

$$p_{\text{noi}}(s) = p_{\text{end}} + (p_{\text{start}} - p_{\text{end}}) \cdot \max\left(0, 1 - \tfrac{s}{S_{\text{total}}}\right) \tag{5}$$

where $s$ is the current training step, $S_{\text{total}}$ is the total training steps. $p_{\text{start}}$ is the initial step sampling probability and $p_{\text{end}}$ is the final step sampling probability. This probability is annealed over the course of training according to a linear decay schedule. Initially, $p_{\text{noi}}$ is high to promote broad exploration of the state space. As training progresses, $p_{\text{noi}}$ is gradually decreased, causing the optimizer to favor the more stable advantage estimates from the raw branch. This allows the policy to first explore and then fine-tune its reasoning based on the original, unperturbed data.

## 4 EXPERIMENTS

### 4.1 EXPERIMENTAL SETUP

We train all models on the dataset MMRL30k (Zhu et al., 2025a) for 200 steps. The training is performed on 8 GPUs, using the AdamW optimizer (Loshchilov and Hutter, 2019) with a learning rate of $1e - 6$ and weight decay of $0.01$. To inject noise into images, we apply Gaussian noise with zero mean and standard deviation $\sigma = 0.4$. When transferring token entropy and visual uncertainty for advantage shaping, we set $\alpha_e = 0.4$, $\beta_e = 2.0$ following the setup of Cheng et al. (2025), and set $\alpha_v = 1.0$, and $\beta_v = 2.0$. The annealed sampling schedule is defined by $p_{\text{start}} = 1.0$ and $p_{\text{end}} = 0$. We adopt a rollout batch size $G$ of 256 and generate $n = 5$ responses per input. The implementation builds on the framework EasyR1 (Zheng et al., 2025).

We conduct direct RL training on top of base models of Qwen2.5-VL-3B and 7B (Bai et al., 2023). The models are trained to generate responses in a structured format, where the reasoning process is enclosed within `<think></think>` tags and the final answer is presented in `\boxed{}`. The reward function is a weighted combination of a format reward and an accuracy reward, with coefficients of $0.1$ and $0.9$, respectively.

We compare VOGUE to two baselines trained under the same setup for a fair comparison: GRPO (Shao et al., 2024) and Pass@k Training (Chen et al., 2025b) (with $k = 4$). For broader context, we also report published results from 7B models: R1-One-Vision-7B (Yang et al., 2025), Vision-R1-7B (Huang et al., 2025), OpenVLThinker-7B (Deng et al., 2025), VLAA-Thinker-7B (Chen et al., 2025a), MM-Eureka-Qwen-7B (Meng et al., 2025), ThinkLite-VL-7B (Wang et al., 2025b), and VL-Rethinker-7B (Wang et al., 2025a). We evaluate pass@1 and pass@4 accuracy on six benchmarks, including MathVerse (Zhang et al., 2024), MathVista (Lu et al., 2023), WeMath (Qiao et al., 2024), HallusionBench (Guan et al., 2024), ChartQA (Masry et al., 2022), and LogicVista (Xiao et al., 2024). These benchmarks span diverse aspects of multimodal reasoning, covering mathematical problem solving, hallucination detection, chart understanding, and logical reasoning. Because Vision-R1-7B used WeMath as training data, we omit its results on that benchmark. For evaluation, we use Qwen2.5-72B-Instruct (Bai et al., 2025) to extract final answers from model responses and assess their correctness against reference answers following prior RLVR work (Su et al., 2025).

### 4.2 MAIN RESULTS

We compare VOGUE against multiple baselines on mathematical and general-domain reasoning. First, we conduct a quantitative evaluation on mathematical reasoning benchmarks, including Math-Verse, MathVista, and WeMath (Table 1). Across Qwen2.5-VL 3B and 7B, VOGUE consistently outperforms the strong RLVR baseline GRPO, demonstrating improved mathematical reasoning. By contrast, Pass@k Training (Chen et al., 2025b), which optimizes the policy with a pass@k-based

Table 1: **Model performance of pass@1 accuracy on diverse visual mathematical reasoning benchmarks.** We compare VOGUE with prior SFT+RL and Zero-RL methods, as well as with GRPO and Pass@k Training baselines on Qwen2.5-VL 3B and 7B models. VOGUE consistently improves over GRPO across both models. And Qwen2.5-VL-7B + VOGUE achieves the strongest average performance across benchmarks.

| Model | MathVerse | MathVista | WeMath | Avg. |
|---|---|---|---|---|
| SFT + RL | | | | |
| R1-One-Vision-7B (Yang et al., 2025) | 42.6 | 62.9 | 60.3 | 55.3 |
| Vision-R1-7B (Huang et al., 2025) | 48.0 | 71.2 | - | - |
| OpenVLThinker-7B (Deng et al., 2025) | 45.4 | 70.0 | 65.8 | 60.4 |
| VLAA-Thinker-7B (Chen et al., 2025a) | 45.8 | 69.3 | 64.3 | 59.8 |
| Zero RL | | | | |
| MM-Eureka-Qwen-7B (Meng et al., 2025) | 47.5 | 71.2 | 65.6 | 61.4 |
| ThinkLite-VL-7B (Wang et al., 2025b) | 44.8 | 73.1 | 64.7 | 60.9 |
| VL-Rethinker-7B (Wang et al., 2025a) | 49.5 | 73.3 | 57.8 | 60.2 |
| Qwen2.5-VL-3B (Bai et al., 2023) | 35.3 | 55.7 | 52.5 | 47.8 |
| + GRPO | 40.6 | 66.4 | 62.8 | 56.6 |
| + Pass@k Training (Chen et al., 2025b) | 30.2 | 56.2 | 45.6 | 44.0 |
| + VOGUE | 42.7 | 68.9 | 63.0 | 58.2 |
| Qwen2.5-VL-7B (Bai et al., 2023) | 43.0 | 66.1 | 62.6 | 57.2 |
| + GRPO | 48.0 | 72.1 | 69.5 | 63.2 |
| + Pass@k Training (Chen et al., 2025b) | 39.6 | 64.6 | 57.0 | 53.7 |
| + VOGUE | **52.1** | **74.2** | **71.1** | **65.8** |

reward, underperforms other methods. **This highlights the unique challenges of multimodal reasoning and provides evidence that exploration strategies designed for text-only settings may not readily transfer.**

To assess generalization beyond mathematical reasoning, we extend our evaluation to a set of broader reasoning tasks using HallusionBench, ChartQA, and LogicVista (Table 2). VOGUE again outperforms GRPO on all three benchmarks, with Qwen2.5-VL-7B + VOGUE achieving the best average performance. The training accuracy reward in Figure 2 further supports this, showing VOGUE's reward curve is consistently above GRPO's for both the 3B and 7B models. **These consistent gains across diverse problem types demonstrate that VOGUE's benefits are robust and not confined to a single domain.**

Finally, we report the pass@4 accuracy across diverse benchmarks in Table 3. While performance in a few cases falls below the base model, a phenomenon well observed in prior works (Yue et al., 2025; Zhu et al., 2025b), VOGUE consistently outperforms the GRPO baseline, and achieves the highest average pass@4 accuracy. **These results confirm VOGUE's effectiveness in promoting exploration.**

Taken together, these results show that VOGUE successfully improves both final-answer accuracy (pass@1) and enhances exploration (pass@4), confirming that guiding exploration with visual uncertainty is an effective strategy for multimodal RLVR.

## 4.3 Ablation Studies

To validate the contribution of each component in VOGUE, we perform a series of ablation studies using the Qwen2.5-VL-7B model. Specifically, we analyze the effects of the visual uncertainty bonus, the token entropy bonus, the annealed strategy, as well as the influence of alternative divergence measures and varying noise levels. We present the training curves in Figure 3 and provide the evaluation results for each setting on six multimodal benchmarks in Appendix A.3.

**Effectiveness of Visual Uncertainty.** We first examine the role of the visual uncertainty by disabling the visual uncertainty term $U_v$ during advantage shaping. As shown in Figure 3a, the resulting learning curve lags behind that of the full VOGUE approach. **This degradation confirms that in-**

Table 2: **Model performance of pass@1 accuracy on diverse visual general-domain reasoning benchmarks.** We compare VOGUE with prior SFT+RL and Zero-RL methods, as well as with GRPO and Pass@k Training baselines on Qwen2.5-VL 3B and 7B models. VOGUE consistently improves over GRPO, with Qwen2.5-VL-7B + VOGUE achieving the strongest average performance across benchmarks.

| Model | HallusionBench | ChartQA | LogicVista | Avg. |
|---|---|---|---|---|
| SFT + RL | | | | |
| R1-One-Vision-7B (Yang et al., 2025) | 67.2 | 78.3 | 45.5 | 63.7 |
| Vision-R1-7B Huang et al. (2025) | 57.8 | 82.7 | 47.8 | 62.7 |
| OpenVLThinker-7B (Deng et al., 2025) | 60.0 | 80.2 | 47.3 | 62.5 |
| VLAA-Thinker-7B (Chen et al., 2025a) | 70.0 | 80.2 | 47.3 | 65.8 |
| Zero RL | | | | |
| MM-Eureka-Qwen-7B (Meng et al., 2025) | 66.4 | 79.9 | 47.3 | 64.5 |
| ThinkLite-VL-7B (Wang et al., 2025b) | 70.9 | 81.4 | **48.9** | 67.0 |
| VL-Rethinker-7B (Wang et al., 2025a) | 69.5 | 81.0 | 48.4 | 66.3 |
| Qwen2.5-VL-3B (Bai et al., 2023) | 61.4 | 73.8 | 33.3 | 56.2 |
| + GRPO | 65.5 | 77.6 | 39.3 | 60.8 |
| + Pass@k Training (Chen et al., 2025b) | 62.3 | 72.6 | 36.8 | 57.2 |
| + VOGUE | 67.0 | 78.1 | 44.0 | 63.0 |
| Qwen2.5-VL-7B (Bai et al., 2023) | 66.9 | 79.8 | 45.5 | 64.1 |
| + GRPO | 68.6 | 81.9 | 42.0 | 64.2 |
| + Pass@k Training (Chen et al., 2025b) | 65.2 | 78.6 | 46.4 | 63.4 |
| + VOGUE | **71.0** | **84.0** | 48.7 | **67.9** |

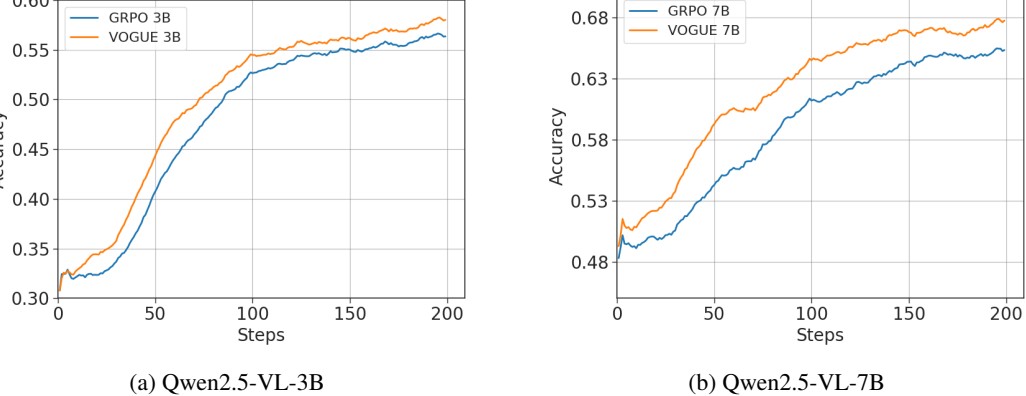

(a) Qwen2.5-VL-3B          (b) Qwen2.5-VL-7B

Figure 2: **Training accuracy rewards of GRPO and VOGUE on Qwen2.5-VL 3B and 7B models**. VOGUE consistently achieves higher rewards than GRPO throughout training.

corporating visual uncertainty is helpful for guiding the agent toward visually uncertain states and thereby enhancing exploration.**

**Effectiveness of Token Entropy.** Next, we analyze the impact of the token entropy by removing the $B_e$ bonus. As illustrated in Figure 3a, performance drops compared to the full approach. Whereas the visual uncertainty targets exploration in the visual state space, token entropy encourages stochasticity in the textual output space. Removing both the visual uncertainty and token entropy bonuses causes a more severe degradation. This is further supported by the results in Table 4 and Table 5 in Appendix A.3. **These results demonstrates that maintaining textual stochasticity is a complementary and beneficial mechanism alongside visually-guided exploration.**

**Effectiveness of Annealed Sampling.** We evaluate the annealed sampling mechanism, which gradually adjusts the probability of selecting the noisy branch versus the raw branch. To isolate its effect, we replace it with a fixed sampling probability of 0.5. The results in Figure 3b show infe-

Table 3: **Model performance of pass@4 accuracy on diverse visual reasoning benchmarks.** On both Qwen2.5-VL 3B and 7B models, VOGUE consistently improves over GRPO and achieves the highest average pass@4 accuracy.

| Model | MathVerse | MathVista | WeMath | ChartQA | LogicVista | Avg. |
|---|---|---|---|---|---|---|
| Qwen2.5-VL-3B (Bai et al., 2023) | **56.5** | 76.8 | 79.0 | 82.0 | **74.1** | 73.7 |
| + GRPO | 53.5 | 77.6 | 82.2 | 81.6 | 65.8 | 72.1 |
| + Pass@k Training (Chen et al., 2025b) | 54.2 | 77.5 | 78.9 | **84.3** | 73.4 | 73.7 |
| + VOGUE | 55.9 | **79.2** | **83.9** | 83.3 | 67.6 | **74.0** |
| Qwen2.5-VL-7B (Bai et al., 2023) | 60.9 | 82.8 | 82.2 | 85.8 | **77.0** | 77.7 |
| + GRPO | 60.2 | 82.5 | 85.0 | 85.1 | 72.1 | 77.0 |
| + Pass@k Training (Chen et al., 2025b) | 60.5 | 80.6 | 80.5 | 85.5 | 75.7 | 76.6 |
| + VOGUE | **61.0** | **83.7** | **86.0** | **87.0** | 72.3 | **78.0** |

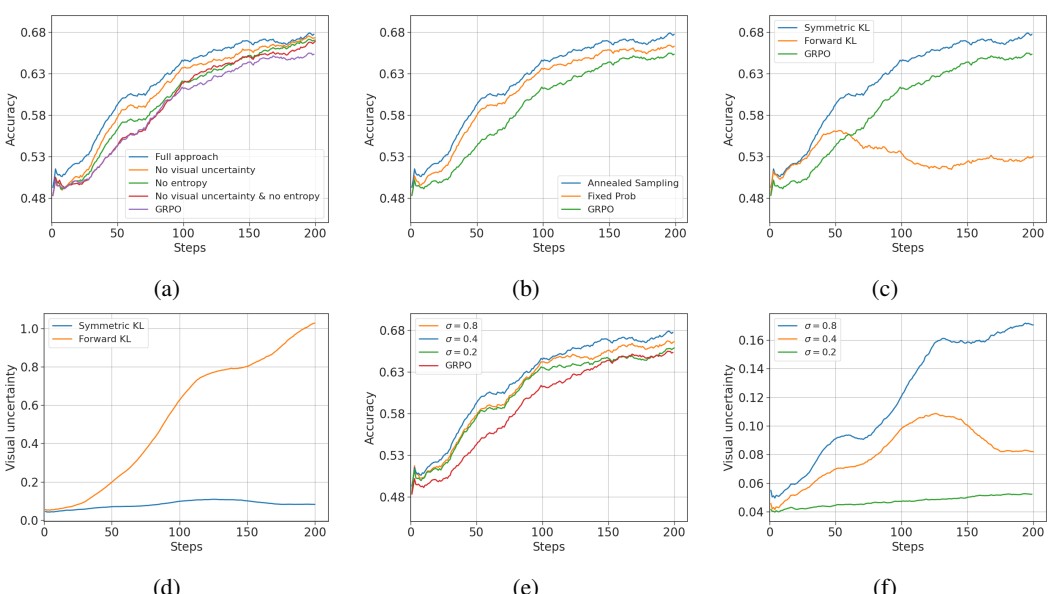

Figure 3: **Ablation studies on the effects of visual uncertainty, token entropy, sampling strategy, divergence measure, and noise level**. (a) Visual uncertainty and token entropy bonuses each improve performance, and together yield the best results. (b) Annealed sampling outperforms fixed sampling, confirming the benefit of dynamically controlling. (c–d) Symmetric KL provides stable gains, while forward KL causes excessive visual uncertainty and degraded accuracy. (e–f) Moderate noise ($\sigma = 0.4$) yields the best accuracy, while low noise limits exploration and high noise increases variance.

rior performance compared to the full annealed strategy. Similar trends are observed in Table 6 and Table 7 in Appendix A.3. **This underscores the benefit of dynamically balancing the trade-off, as early-stage exploration requires more noisy-branch updates while later-stage convergence benefits from the stability of the raw branch.**

**Alternative Divergence Measures.** To estimate visual uncertainty, we experiment with a forward KL formulation instead of symmetric KL. However, as shown in Figure 3c, the forward KL divergence leads to unstable training with accuracy declining. This occurs because forward KL encourages the model to diverge excessively, which is reflected in Figure 3d, where the resulting visual uncertainty becomes excessively large. Results in Table 8 and Table 9 in Appendix A.3 further confirm this finding. **These results validate our choice of symmetric KL for promoting exploration while maintaining training stability.**

**Different Noise Levels.** Finally, We evaluate VOGUE under different noise injection levels by varying the standard deviation of the Gaussian perturbation in the noisy branch ($\sigma \in 0.2, 0.4, 0.8$). The results in Figure 3e indicate that moderate noise ($\sigma = 0.4$) provides the best accuracy. Low noise ($\sigma = 0.2$) yields insufficient visual exploration, while high noise ($\sigma = 0.8$) introduces excessive variance. Results in Table 10 and Table 11 in Appendix A.3 align with this observation. **This shows**

**that while noise is essential for quantifying uncertainty, an appropriate level is needed to avoid both insufficient exploration and excessive variance.**

**Taken together, these ablation studies confirm that each component of VOGUE plays a distinct and complementary role.** The visual uncertainty bonus directs exploration in the visual space, token entropy sustains diversity in the textual output space, and annealed sampling adaptively balances noisy and raw branches. Our analysis of divergence measures and noise levels further validates the design choices that make VOGUE effective and stable.

## 5 RELATED WORK

**Exploration in Text-Based Reasoning.** Recent work on text-only RLVR has begun to tackle exploration explicitly. Complementary to outcome rewards, i-MENTOR (Gao et al., 2025) incorporates trajectory-aware intrinsic signals and dynamic reward scaling to improve exploration for LLM reasoning. Replay-style methods, such as Retrospective Replay (Dou et al., 2025), revisit promising early states to prevent exploration decay later in training. Others have proposed outcome-based exploration schemes (Song et al., 2025) or have leveraged process rewards for more granular guidance (Setlur et al., 2024), though reliably scoring intermediate steps remains a challenge. Additionally, upweighting negative-sample reinforcement has been shown to mitigate diversity collapse and improve Pass@k (Zhu et al., 2025b).

**Multimodal RLVR.** RLVR has been increasingly applied to enhance the reasoning capabilities of multimodal models. Yang et al. (2025) extended language reasoning with visual inputs, improving visual question answering, while Huang et al. (2025) employed vision-grounded prompts to enhance multi-step reasoning. Deng et al. (2025) leveraged large-scale visual instruction tuning for improved cross-modal generalization, and Chen et al. (2025a) unified visual and textual signals in a policy-learning framework. Meng et al. (2025) introduced hierarchical visual abstractions for RL-guided multimodal planning and Wang et al. (2025a) iteratively refined answers through visual reasoning and reflection. More recently, Li et al. (2025b) proposed a self-rewarding framework that decomposes reasoning into visual perception and language reasoning, using self-generated rewards to improve visual reasoning and reduce hallucinations.

Despite these advances, the critical balance of exploration and exploitation remains underexplored in multimodal RLVR. Prior works typically focus on pass@1 accuracy Liu et al. (2025c); Zhu et al. (2025a). Some studies note insufficient exploration in RL algorithms such as GRPO (Shao et al., 2024), proposing dynamic KL strategies (Liu et al., 2025b) or rule-based process rewards (Zhang et al., 2025), yet they do not explicitly encourage exploration. More recently, Pass@k Training (Chen et al., 2025b) used pass@k as the training reward and analyzed exploration–exploitation trade-off, but its focus was primarily on text-based reasoning, offering limited insights for multimodal scenarios.

In contrast, VOGUE addresses this gap by coupling exploration to quantified visual uncertainty and, to our knowledge, is among the first modality-aware exploration frameworks for RLVR. Moreover, VOGUE is complementary to other language-side exploration strategies (e.g., temperature scheduling (Liao et al., 2025), KL regularization (Liu et al., 2025a), and output-level diversity/novelty bonuses (Li et al., 2025a)), which have shown benefits mainly in text RL and can be combined for further gains.

## 6 CONCLUSIONS

We introduce VOGUE, a visual-uncertainty–guided exploration method for multimodal RLVR that treats the image as a stochastic context. By quantifying sensitivity to semantics-preserving perturbations and shaping advantages with uncertainty- and entropy-based bonuses under an annealed sampling schedule, VOGUE couples exploration directly to visual uncertainty while maintaining stable optimization. Compared to the strong RLVR baseline GRPO, VOGUE achieves consistent improvements in both pass@1 and pass@4 accuracy across diverse benchmarks, including mathematical problem solving, hallucination detection, chart understanding, and logical reasoning. These results highlight the effectiveness of VOGUE in enhancing multimodal reasoning. For a discussion on future work, please see Appendix A.4.

## 7 REPRODUCIBILITY STATEMENT

To ensure reproducibility, we provide detailed descriptions of all training setups in Section 4.1 of experimental setup. We list the system prompts used for training in Appendix A.2. The dataset used is publicly available, and we will release the code upon publication.

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

# A APPENDIX

## A.1 THE USE OF LLMS

We use LLMs to polish paper writing and refine grammar of the manuscript.

## A.2 PROMPT TEMPLATES

> **System Prompt**
>
> You FIRST think about the reasoning process as an internal monologue and then provide the final answer. The reasoning process MUST be enclosed within `<think></think>` tags. The final answer MUST be put in `\boxed{}`.

## A.3 ABLATION STUDIES

To assess the contribution of each component in VOGUE, we conduct a series of ablation studies using the Qwen2.5-VL-7B model. We examine the effects of the visual uncertainty bonus, the token entropy bonus, the annealed sampling strategy, as well as the impact of alternative divergence measures and varying noise levels. Evaluation is performed across six multimodal benchmarks, including MathVerse (Zhang et al., 2024), MathVista (Lu et al., 2023), WeMath (Qiao et al., 2024), HallusionBench (Guan et al., 2024), ChartQA (Masry et al., 2022), and LogicVista (Xiao et al., 2024). These benchmarks together cover diverse aspects of multimodal reasoning, including mathematical problem solving, hallucination detection, chart interpretation, and logical reasoning. We report pass@1 accuracy in the following tables for each ablation setting.

Table 4: **Pass@1 accuracy on mathematical reasoning benchmarks testing the effectiveness of visual uncertainty and token entropy.** Removing either visual uncertainty or token entropy reduces performance, while removing both leads to a larger drop. This confirms that both components are effective for enhancing exploration and improving performance.

| Approach | MathVerse | MathVista | WeMath | Avg. |
|---|---|---|---|---|
| GRPO | 48.0 | 72.1 | 69.5 | 63.2 |
| VOGUE | | | | |
| Full approach | 52.1 | 74.2 | 71.1 | 65.8 |
| − Visual uncertainty | 48.3 | 73.6 | 70.3 | 64.1 |
| − Entropy | 48.6 | 73.5 | 70.8 | 64.3 |
| − Visual uncertainty & entropy | 48.3 | 73.1 | 68.5 | 63.3 |

Table 5: **Pass@1 accuracy on general-domain reasoning benchmarks testing the effectiveness of visual uncertainty and token entropy.** Removing either visual uncertainty or token entropy reduces performance, while removing both leads to a larger drop. This demonstrates that both components enhance exploration and thereby improve performance.

| Approach | HallusionBench | ChartQA | LogicVista | Avg. |
|---|---|---|---|---|
| GRPO | 68.6 | 81.9 | 42.0 | 64.2 |
| VOGUE | | | | |
| Full approach | 71.0 | 84.0 | 48.7 | 67.9 |
| − Visual uncertainty | 69.7 | 83.4 | 47.8 | 66.9 |
| − Entropy | 70.2 | 82.4 | 47.8 | 66.8 |
| − Visual uncertainty & entropy | 69.2 | 82.1 | 46.4 | 65.9 |

Table 6: **Pass@1 accuracy on mathematical reasoning benchmarks testing the effectiveness of annealed sampling.** Using fixed-probability sampling yields lower performance compared to annealed sampling, underscoring the benefit of dynamically adjusting sampling probability.

| Approach | MathVerse | MathVista | WeMath | Avg. |
|---|---|---|---|---|
| GRPO | 48.0 | 72.1 | 69.5 | 63.2 |
| VOGUE | | | | |
| Annealed Sampling | 52.1 | 74.2 | 71.1 | 65.8 |
| Fixed Prob | 48.5 | 73.6 | 67.8 | 63.3 |

Table 7: **Pass@1 accuracy on general-domain reasoning benchmarks testing the effectiveness of annealed sampling.** Fixed-probability sampling yields lower performance than annealed sampling, highlighting the benefit of dynamically controlling the sampling probability.

| Approach | HallusionBench | ChartQA | LogicVista | Avg. |
|---|---|---|---|---|
| GRPO | 68.6 | 81.9 | 42.0 | 64.2 |
| VOGUE | | | | |
| Annealed Sampling | 71.0 | 84.0 | 48.7 | 67.9 |
| Fixed Prob | 69.9 | 82.6 | 46.9 | 66.5 |

A.4 DISCUSSIONS AND FUTURE WORK

A key strength of VOGUE is its modularity and practical design. While implemented here within GRPO, its core mechanism is readily adaptable to other policy gradient methods and requires no additional supervision, making it a practical, drop-in enhancement. This practicality extends to its computational profile: VOGUE's substantial performance gains are achieved with a modest 20% overhead (4.95 vs. 4.12 minutes per step), a trade-off that is highly efficient compared to naive online augmentation (i.e., treating each augmented view as an independent training sample). By successfully pioneering the use of visual uncertainty, this work suggests a promising direction for future exploration into more complex, cross-modal uncertainty schemes. Future research could extend this framework to adaptively perturb both visual and textual inputs, potentially capturing richer uncertainty landscapes and further strengthening reasoning agents.

Table 8: **Pass@1 accuracy on mathematical reasoning benchmarks with alternative divergence measures.** The results validate our choice of symmetric KL, which promotes exploration while maintaining training stability, whereas forward KL causes excessive divergence and degrades performance.

| Approach | MathVerse | MathVista | WeMath | Avg. |
|---|---|---|---|---|
| GRPO | 48.0 | 72.1 | 69.5 | 63.2 |
| VOGUE | | | | |
| Symmetric KL | 52.1 | 74.2 | 71.1 | 65.8 |
| Forward KL | 39.4 | 70.7 | 56.1 | 55.4 |

Table 9: **Pass@1 accuracy on general-domain reasoning benchmarks with alternative divergence measures.** The results validate symmetric KL as it promotes exploration while maintaining training stability, whereas forward KL causes excessive divergence and degrades performance.

| Approach | HallusionBench | ChartQA | LogicVista | Avg. |
|---|---|---|---|---|
| GRPO | 68.6 | 81.9 | 42.0 | 64.2 |
| VOGUE | | | | |
| Symmetric KL | 71.0 | 84.0 | 48.7 | 67.9 |
| Forward KL | 67.5 | 80.3 | 45.1 | 64.3 |

Table 10: **Pass@1 accuracy on mathematical reasoning benchmarks with different noise levels.** Moderate noise ($\sigma = 0.4$) yields the best accuracy.

| Approach | MathVerse | MathVista | WeMath | Avg. |
|---|---|---|---|---|
| GRPO | 48.0 | 72.1 | 69.5 | 63.2 |
| VOGUE | | | | |
| $\sigma = 0.2$ | 48.4 | 74.0 | 68.8 | 63.7 |
| $\sigma = 0.4$ | 52.1 | 74.2 | 71.1 | 65.8 |
| $\sigma = 0.8$ | 49.2 | 73.5 | 66.8 | 63.2 |

Table 11: **Pass@1 accuracy on general-domain reasoning benchmarks with different noise levels.** Moderate noise ($\sigma = 0.4$) yields the best accuracy.

| Approach | HallusionBench | ChartQA | LogicVista | Avg. |
|---|---|---|---|---|
| GRPO | 68.6 | 81.9 | 42.0 | 64.2 |
| VOGUE | | | | |
| $\sigma = 0.2$ | 69.4 | 81.9 | 45.3 | 65.5 |
| $\sigma = 0.4$ | 71.0 | 84.0 | 48.7 | 67.9 |
| $\sigma = 0.8$ | 70.4 | 82.9 | 46.2 | 66.5 |

