# OpenReview forum: "VOGUE: Guiding Exploration with Visual Uncertainty Improves Multimodal Reasoning"
_ICLR.cc/2026/Conference — ICLR 2026 Conference Withdrawn Submission_

### Official Review · Reviewer_avNu · 2025-10-30

**Soundness:** 2
**Presentation:** 3
**Contribution:** 1
**Rating:** 2
**Confidence:** 5

**Summary:**

VOGUE introduces a dual-branch RL fine-tuning scheme for MLLMs: a raw image branch and a noisy image branch. A token-level symmetric KL between branches yields a visual-uncertainty signal that shapes the noisy-branch advantage; an entropy bonus and an annealed branch-sampling schedule further balance exploration vs. exploitation. Experiments on Qwen2.5‑VL‑3B/7B over six benchmarks report pass@1/pass@4 gains; ablations cover divergence choice and noise level.

**Strengths:**

- Simple, practical mechanism. Clear, implementable recipe: dual forward passes, capped uncertainty/entropy bonuses, and linear annealing of the noisy‑branch sampling probability.
- Consistent average gains. Pass@1 improves across math and general‑domain benchmarks for both 3B and 7B models; pass@4 averages also rise vs. GRPO.

**Weaknesses:**

1) The paper does not cite or discuss NoisyRollout [1], despite very similar high‑level motivation, introducing controlled visual noise during RL to encourage exploration. NoisyRollout mixes clean + moderately distorted rollouts and employs noise annealing; VOGUE uses symmetric‑KL advantage shaping with annealed branch sampling. A head‑to‑head is essential to establish incremental value beyond noise‑driven exploration.

2) Experiments run for 200 steps on Qwen2.5‑VL‑3B/7B without multi‑seed reporting, error bars, or confidence intervals; several pass@4 entries are below the base model (e.g., 3B on LogicVista: 74.1 base vs. 67.6 VOGUE in Table 3). Claims around mitigating “exploration decay” would be stronger with longer training and variance analysis.

3) Section 3.1 defines stochastic augmentation function T to include flips/rotations/jitter/Gaussian noise, but §4.1 states training “apply Gaussian noise with σ = 0.4” (no mention of flips/rotations/jitter in the actual runs). The paper should clarify whether non‑Gaussian transforms were used in any experiments; if not, method and experiments should be reconciled.

4) The paper does not sweep key caps ((alpha_v,beta_v)), ((alpha_e,beta_e) or annealing endpoints; only fixed values are reported (α_v = 1.0, β_v = 2.0; α_e = 0.4, β_e = 2.0). This limits insight into stability/sensitivity.

5) Using data augmentation is a common practice in reinforcement learning and has been widely explored, especially in visual RL, so employing it here isn’t particularly novel (which I think is perfectly fine). Works such as DrQ [2] and RAD [3] should be at least cited and discussed in the related work section.

[1] NoisyRollout: Reinforcing Visual Reasoning with Data Augmentation. In Advances in Neural Information Processing Systems, 2025.

[2] Image Augmentation Is All You Need: Regularizing Deep Reinforcement Learning from Pixels. In 9th International Conference on Learning Representations, 2021.

[3] Reinforcement Learning with Augmented Data. In Advances in Neural Information Processing Systems, 2020.

**Questions:**

1. Please cite NoisyRollout and provide a controlled comparison (same model/data/steps/compute). Where does symmetric‑KL advantage shaping + annealed branch sampling*outperform clean+noisy rollout mixing + noise annealing? A head-to-head comparison is prefered.
2. Clarify whether training used only Gaussian noise or also flips/rotations/jitter as described in section 3.1. If only Gaussian noise is used, please adjust section 3.1 accordingly; if broader transforms are used, specify when and how.
3. Provide multi‑seed CIs for Tables 1–3 and add error bands to training curves.
4. Report sweeps for alpha_v,beta_v, alpha_e,beta_e, and p_{start},p_{end} to understand stability and the exploration‑exploitation trade‑off.
5. Have you examined gradient norms or variance across raw/noisy branches? Do noisy gradients consistently guide the policy toward visually ambiguous yet semantically relevant states, or do they sometimes cause destructive updates? An empirical gradient-analysis would help.

---

### Official Review · Reviewer_VGJ7 · 2025-10-31

**Soundness:** 2
**Presentation:** 2
**Contribution:** 1
**Rating:** 2
**Confidence:** 5

**Summary:**

This paper proposes VOGUE, a reinforcement learning framework that introduces visual uncertainty to guide exploration for multimodal large language models (MLLMs) during RL with verifiable rewards (RLVR) training. The method perturbs input images with semantics-preserving noise, measures the model’s sensitivity via a symmetric KL divergence between response distributions, and incorporates this “visual uncertainty” as an exploration bonus. It further introduces a token-level entropy bonus and an annealed dual-branch sampling strategy. Experiments on several visual reasoning benchmarks (MathVerse, MathVista, WeMath, HallusionBench, ChartQA, LogicVista) show consistent but moderate performance improvements over GRPO baselines.

**Strengths:**

- The paper is clearly structured and easy to follow.
- VOGUE demonstrates consistent empirical improvements across several visual reasoning benchmarks.
- The ablation results support the contribution of each component, indicating solid experimental design.

**Weaknesses:**

- The main contribution—introducing visual uncertainty and entropy bonus into RLVR training for vision-language models—is conceptually aligned with many existing works [1,2,3,4]. These ideas have been discussed widely in the literature, so the originality is limited.
- The visual uncertainty component overlaps strongly with NoisyRollout [2] in both motivation (enhancing policy exploration), implementation (annealed sampling, branched rollout strategy), and even empirical findings (moderate noise level works best). However, the paper makes no reference or discussion to this related work.
- The baseline results raise questions. Some baselines seem weaker than expected; for example, VL-Rethinker’s reported accuracy on MathVerse (49.5) is substantially lower than in its original paper (54.2), and its WeMath result is also much worse than other weak baselines. It would help if the authors clarified evaluation setups and re-implemented baselines consistently.
- The paper overlooks an ablation study: increasing decoding temperature is also a standard and effective way to enhance exploration in RL-based training. The lack of comparison or discussion on this aspect weakens the completeness of the exploration analysis.
- In RLVR and more broadly in RL for LLMs, entropy is usually treated as an observational metric (to monitor diversity) or a regularization term (to stabilize training), rather than being directly optimized as an explicit objective. Framing it as a primary optimization target, as done in this paper, may raise conceptual questions and requires stronger theoretical justification.

[1]. NoisyRollout: Reinforcing Visual Reasoning with Data Augmentation. NeurIPS 2025.

[2]. R1-ShareVL: Incentivizing Reasoning Capability of Multimodal Large Language Models via Share-GRPO. NeurIPS 2025.

[3]. GTPO and GRPO-S: Token and Sequence-Level Reward Shaping with Policy Entropy. Arxiv 2025.

[4]. Beyond the 80/20 Rule: High-Entropy Minority Tokens Drive Effective Reinforcement Learning for LLM Reasoning. NeuIPS 2025.

**Questions:**

See above

---

### Official Review · Reviewer_x3Fd · 2025-10-31

**Soundness:** 2
**Presentation:** 3
**Contribution:** 3
**Rating:** 4
**Confidence:** 3

**Summary:**

This paper presents VOGUE (Visual-Uncertainty–Guided Exploration), a novel reinforcement learning framework that enhances reasoning in multimodal large language models (MLLMs) by addressing their limited exploration capabilities. Unlike existing methods that treat visual inputs as fixed conditions, VOGUE models them as stochastic contexts and measures the model’s sensitivity to visual perturbations through symmetric KL divergence between a raw and a noisy input branch. This uncertainty signal drives exploration via an uncertainty-proportional bonus, complemented by a token-entropy bonus and an annealed sampling schedule to balance exploration and exploitation. Integrated into GRPO and evaluated on Qwen2.5-VL-3B/7B, VOGUE achieves average improvements of 2.6% on visual math and 3.7% on general reasoning benchmarks, while mitigating exploration decay—demonstrating that leveraging visual uncertainty can significantly strengthen multimodal reasoning performance.

**Strengths:**

1. The paper effectively leverages visual uncertainty as a novel exploration mechanism in multimodal large language models, offering a principled way to enhance reasoning beyond existing approaches.
2. It demonstrates strong and consistent empirical improvements across benchmarks, showing that the proposed VOGUE framework boosts both exploitation and exploration performance over competitive baselines.
3. The paper is well-written, presenting a straight-forward idea with conceptual clarity and a logical narrative that is easy to follow.

**Weaknesses:**

1. The method relies heavily on the assumption that the visual perturbations used in the noisy branch preserve the semantics of the original image. However, in complex multimodal datasets (e.g., math diagrams, charts), even small transformations can inadvertently alter meaning—introducing noise that is not truly semantic-preserving. This could introduce spurious uncertainty rather than useful exploratory signals.
2. While VOGUE advances visual-side exploration, it does not explicitly account for uncertainty in the textual modality, even though textual prompts or reasoning steps could also carry ambiguity. This creates a potential imbalance in the exploration space, possibly neglecting useful trajectories in the language domain.

**Questions:**

1. I'm curious whether the proposed idea can be extended to jointly model and explore uncertainty in both the visual and textual inputs.
2. The method uses an annealed sampling schedule that gradually transitions from noisy to raw image branches. Is this schedule universally optimal across all tasks (e.g., hallucination detection vs. math reasoning), or would task-specific tuning of annealing parameters further improve performance?

---

### Official Review · Reviewer_8zKw · 2025-11-04

**Soundness:** 3
**Presentation:** 2
**Contribution:** 2
**Rating:** 4
**Confidence:** 2

**Summary:**

This paper introduces VOGUE (Visual-Uncertainty–Guided Exploration), a method for improving multimodal reinforcement learning by shifting exploration from the output (text) to the input (visual) space. The approach uses a dual-branch architecture: a "raw" branch processing original images and a "noisy" branch processing perturbed images. Visual uncertainty is quantified via symmetric KL divergence between branches and used to shape advantages through an uncertainty-proportional bonus, combined with token entropy bonuses and an annealed sampling schedule. Implemented within GRPO on Qwen2.5-VL-3B/7B models, VOGUE achieves average gains of 2.6% on math benchmarks and 3.7% on general reasoning benchmarks in pass@1 accuracy, while also improving pass@4 performance.

**Strengths:**

- The paper presents a novel perspective by grounding exploration in visual input uncertainty rather than textual output diversity, which represents a valuable insight for multimodal RLVR. The dual-branch architecture with symmetric KL divergence provides a principled way to quantify visual uncertainty and guide exploration.
- Empirical validation is comprehensive and convincing, spanning six diverse benchmarks (MathVerse, MathVista, WeMath, HallusionBench, ChartQA, LogicVista) where VOGUE demonstrates consistent improvements in both pass@1 metrics for exploitation and pass@4 metrics for exploration.
- The ablation studies are thorough and well-designed, systematically validating each component's contribution including visual uncertainty, entropy bonus, annealed sampling, divergence measures, and noise levels, with clear explanations supporting the design choices.

**Weaknesses:**

- The augmentation strategy is simplistic, relying only on basic transformations such as flips, rotations, color jitter, and Gaussian noise. The paper lacks exploration of more sophisticated, task-aware augmentations that could better capture semantic-preserving variations specifically relevant to reasoning tasks.
- Computational cost analysis is insufficient beyond reporting a 20% time overhead. Critical details are missing regarding memory footprint, scalability to larger models beyond 7B parameters, and whether the dual-branch forward pass creates bottlenecks for practical deployment at scale.

**Questions:**

- Why does Pass@k Training fail so dramatically in multimodal settings compared to text-only domains? Can you provide deeper analysis of what makes exploration fundamentally different in multimodal vs. text-only RLVR?
- Can you visualize how visual uncertainty evolves during training and provide per-benchmark analysis? This would reveal whether the model progressively resolves visual ambiguities or if uncertainty patterns differ across task types.
- Does VOGUE's effectiveness vary across visual content types (charts vs. natural images vs. mathematical diagrams)?

---

### Note · Authors · 2025-12-17

I have read and agree with the venue's withdrawal policy on behalf of myself and my co-authors.